# Seasonal Movement Patterns of Urban Domestic Cats Living on the Edge in an African City

**DOI:** 10.3390/ani13061013

**Published:** 2023-03-10

**Authors:** Robert E. Simmons, Colleen L. Seymour, Sharon T. George, Koebraa Peters, Frances Morling, M. Justin O’Riain

**Affiliations:** 1DST/NRF Centre of Excellence, FitzPatrick Institute, University of Cape Town, Cape Town 7701, South Africa; 2Kirstenbosch Research Centre, South African National Biodiversity Institute, Claremont 7735, South Africa; 3Department of Conservation and Marine Sciences, Cape Peninsula University of Technology, Cape Town 8000, South Africa; 4Independent Researcher, 16 Taranga Street, Auckland 0627, New Zealand; 5Department of Biological Sciences, Institute for Communities and Wildlife in Africa, University of Cape Town, Cape Town 7701, South Africa

**Keywords:** free-roaming cats, GPS tracking, home range, 95% kernel, protected area buffers, season

## Abstract

**Simple Summary:**

How domestic cats use open spaces around their homes is unstudied in Africa, and this has conservation implications given their high rate of predation on native prey. We GPS-tracked a sample of cats in summer and winter to understand habitat and area use and distances travelled. Since Cape Town surrounds the Table Mountain National Park (TMNP), we also determined how often cats ventured into protected areas. A far greater proportion of cats (59% of 78) returned prey home in summer than winter (30% of 27), and summer ranges were significantly greater and ca. three-fold larger than those in winter (3.00 ha vs. 0.87 ha). Urban-edge (UE) cats travelled up to 850 m from their homes and both urban (U) and UE cats entered natural habitat. All seven GPS-collared UE cats (and one of seven U) ventured into protected areas in summer and two of four UE (and two of five U) cats did so in winter. Thus, our data suggest that cats may regularly hunt in protected areas, especially in summer. Yet they may also limit the time spent in such habitats due to predation risk from meso-carnivores. The threat to biodiversity in protected areas by owned cats necessitates further layers of protection. Cat-free buffers of ~600 m, based on the average movements reported here, may reduce domestic cat predation in protected areas.

**Abstract:**

Domestic cats (*Felis catus*) are amongst the most destructive invasive vertebrates globally, depredating billions of native animals annually. The size and seasonal variation of their geographical “footprint” is key to understanding their effects on wildlife, particularly if they live near conservation areas. Here we report the first GPS-tracking studies of free-roaming owned cats in the city of Cape Town, South Africa. A total of 23 cats was tracked (14 cats in summer, 9 in winter) using miniature (22 g) GPS locators in 2010–2011. In summer, all cats living on the urban-edge (UE: *n* = 7) made extensive use of protected areas, while only one of seven urban (U) cats (>150 m from the edge) did so. In winter two of four UE and two of five U cats entered protected areas. Home ranges (95% kernel density estimates) were significantly larger in summer (3.00 ± 1.23 ha) than winter (0.87 ± 0.25 ha) and cats ventured further from their homes in summer (maximum 849 m) than in winter (max 298 m). The predation risk posed by caracal (*Caracal caracal*) may limit the time cats spend in protected areas, but our results suggest that cat buffers around conservation areas should be at least ~600 m wide to reduce impacts to native fauna.

## 1. Introduction

As major contributors to global biodiversity loss, domestic cats (*Felis catus*) have been studied on every continent where they occur (e.g., [1,2,3,4,5]). The world’s estimated 600 million cats [6] kill billions of birds and mammals annually [2,5], with particularly high tolls for reptiles and amphibians in some regions [1,7,8]. Even these figures likely do not capture the true predation rates, given that most studies rely on prey returned home, under-estimating actual numbers taken by between 4.3 [9] and 5.6-fold [8]. Prey returns also bias perceptions of prey composition, as KittyCam studies reveal mammals and birds are more likely to be returned home than reptiles [8,9].

Free-roaming owned cats in urban areas create an almost continuous “catscape” with few refugia for native fauna [10]. Quantifying the distribution, movement patterns and impacts of domestic cats has been achieved in many parts of the world including Australia, Canada, Denmark, New Zealand, Norway, UK and the USA, [3,10] with only a single study from Africa [8] on predation rates.

Given their impacts on biodiversity through predation and creation of a landscape of fear [11,12], it is essential to know how far cats roam into natural and protected spaces both within and adjacent to the urban matrix. Data on these movement patterns can help inform policy that seeks to prevent or reduce the impacts of free-roaming domestic cats on biodiversity [13], through, for example, the establishment of cat-free buffer zones (~400–600 m wide) around protected areas by banning cat ownership or restricting cats to owners’ properties.

Previous studies have found that owned cats (distinguished from un-owned or feral cats) hunt more during warm months [8,9,14], leading to the prediction that cats may travel further and spend more time away from home in the summer than winter and that the “catscape” may expand and contract over the year. Here, we report on the first GPS-tracking studies of domestic cats in an African city in summer and winter to determine (i) daily distances moved, home range size and habitat use in summer and winter; (ii) whether home range size differed between cats living in the urban matrix (>150 m from green spaces) versus urban-edge (<150 m from green spaces) areas; (iii) roaming during the day vs. night; and (iv) whether, and to what extent, cats made use of green spaces and protected areas within and adjacent to the cadastral boundaries of the City of Cape Town, South Africa.

## 2. Materials and Methods

### 2.1. Study Areas

We tracked the movements of 23 owned cats, 12 from homes completely surrounded by the urban matrix of houses and roads >150 m from the nearest large natural fragments, hereafter referred to as “urban” (U), and 11 from homes within 150 m of the nearest, large natural fragments including protected areas, hereafter “urban-edge” (UE) cats, within Greater Cape Town. Fourteen of these cats were tracked in summer and nine in winter. When this study was initiated (September 2009), there were no published GPS data on cat roaming distances. We therefore selected 150 m as the cut-off distance from natural areas between U and UE cats based on our knowledge of cat behaviour. This is slightly smaller than the average now being found in many studies, but the delineation into U or UE is justified given that our “urban” cats were surrounded by houses and roads, while our “urban edge” cats were all within sight/sound and easy access of natural areas. This is also similar to a 98 m radius circle representing a circular 3 ha home range and thus biologically meaningful in hindsight. It also allows us to determine the distance an urban cat is prepared to go to penetrate a natural fragment.

Cape Town lies in the temperate south-western corner of South Africa (33°55’ S, 18°25’ E) (Figure 1) and is the second most populous city in South Africa at 4.8 million people (https://worldpopulationreview.com/world-cities/cape-town-population (accessed on 1 June 2020)). At its centre is the iconic Table Mountain National Park (TMNP), which extends 49.5 km from Signal Hill (33°55′00″ S, E18°24′16″ E) in the north to Cape Point (34°20′45″ S, 18°29′14″ E) in the south, rising to >1 km at its highest point and covering 265 km² [15]. TMNP is part of the Cape Floristic Region, a biodiversity hotspot that supports a wide range of both flora and fauna endemic to the region [16]. Biodiversity in this region is threatened by urbanization and fragmentation, the spread of invasive alien species, climate change and pollution [17,18]. TMNP is not populated but is almost completely surrounded by suburbs and the Atlantic Ocean (Figure 1).

### 2.2. Study Cats

We tracked cats in two studies, the first in the austral spring/summer (October 2009–January 2010) and the second in the austral winter to early spring (June–September 2011), in which 14 and 9 free-roaming cats were tracked, respectively. The 14 cats in the summer study were from 11 suburbs scattered evenly around TMNP, with seven cats considered UE and seven considered urban (U in Figure 1). The nine cats in the winter study were from two suburbs that were a subset of the original 11, on the edge of TMNP (Glencairn, and Welcome Glen: Figure 1), of which four were UE cats and five were U. The suburbs used in the winter study were representative of the suburbs used for the larger summer study as they comprised an urban matrix and natural habitat including a municipal green space (the Glencairn Wetland Reserve) and TMNP. The mean age (±SD) of U cats in the summer sample was 7.6 ± 7.2 years and for UE cats was 3.8 ± 4.0 years. In the winter sample the mean age of U cats was 8.2 ± 5.1 years and 5.7 ± 1.8 years for UE cats. The sex ratio was 6 males to 8 females in summer and 4 males to 5 females in the winter.

In addition to GPS tracking our study cats, we simultaneously asked cat owners to record all prey items returned by 78 cats in the summer study and 27 cats in the winter study (details in [8,19,20]). To recruit “summer” cats, we distributed 600 questionnaires to cat owners in the suburbs surrounding TMNP (Appendix A). These asked for details of the cat and provided criteria for collecting prey returns over 6–10 weeks. Most questionnaires (350) were distributed through veterinary clinics, with 200 delivered directly to households through door-to-door surveys in two suburbs—Glencairn Heights and Hout Bay (Figure 1). Cat owners were given a summary of the study and asked if they were willing to volunteer their cats for the GPS study. Similar methods were used to recruit cats in the winter sample, employing 64 questionnaire surveys (Appendix A) as before, coupled with door-to-door surveys of 204 houses.

Commercially available cat collars were adapted to take a soft leather pouch, tailor-made to house the GPS tracker (CatTrack^TM^, http://www.mr-lee-catcam.de/index.html (accessed on 13 September 2009)) with dimensions 4.4 × 2.7 × 1.3 cm and weighing 22 g. A metal counterweight (28 g) was attached to the opposite side of the collar to maintain the GPS unit pointing skyward. The total weight of the GPS and counterweight was thus 50 g, which is <1.4% of the lightest cat (3.5 kg) in our sample and well within the maximum percentage weight of <2% of body mass [21]. This was approved by the University of Cape Town ethics committee.

Data collection only commenced after two days following attachment of the collar to allow the cats to habituate to the novel device. The owners were asked to monitor for any signs of discomfort linked to the collars and/or any changes to their general behavior. Only cats for which the collar did not seem to be an irritation, as gauged by the owners, were used in this study. Those observed pawing at the collars or trying to remove them in other ways were omitted from the study. Most cat owners preferred to remove the collar when their cats were at home during the day to charge the batteries and reduce the amount of time that the collar was worn. The owners were careful not to allow the cats outside without the collars during the study. The collared cats actively hunted and ventured into natural habitat, suggesting that the devices did not impede their movement patterns or activities, similar to other tracking studies [3,14]. Furthermore, cats collared with the larger KittyCams had similar predation rates with and without the devices [8] and no adverse effects were reported by the owners.

The GPS loggers recorded a location every 2 min, allowing an average battery life of 100 h. Of the 14 cats in our summer study, two were tracked for 120 h (5 days), nine for 168 h (7 days) and three for 240 h (10 days). In our winter study, eight of the nine cats provided 7 days of tracking; one cat with just 3 days tracking data was omitted from most further analysis. Five to six days of tracking is typically considered sufficient to reach an asymptote in home-range size for most cats over a broad range of study areas [3,14]. However, inclement weather or other factors may mean that the full home range size is not reached in 5–6 days; thus, the mean home ranges given here represent minimum estimates. Collars were retrieved after seven days and data downloaded from the GPS tracking devices.

The area used by each cat was calculated using the “Animal Movement Extension” in ArcView. GIS 3.3 minimum convex polygons (MCP) and fixed kernel methods were used to calculate the area used by each cat. The total area used was defined using MCP 100% confidence limits rather than 95% confidence limits, because 95% MCPs exclude infrequently visited sites [22]. Fixed kernels were generated to identify regions of differential space use (at the 95% level). The smoothing parameter or bandwidth (*h*) ranged from 0.22 to 0.29. The resultant utilization distribution density was converted into probability contours to produce visual displays of the range size and shape (Figure 2). The total area used was defined as the smallest area under the utilization distribution that encompassed 95% of the points, and the core areas were defined as the smallest area encompassing 50% of all points. Given that one aim of the study was to determine whether cats use protected areas and other natural habitats, it was crucial to include these outliers in our assessments.

Habitat use and selection were evaluated for both U and UE cats using selection ratios [11,23]. Three main habitats were identified: (i) natural (i.e., protected areas and other large patches of natural vegetation), (ii) urban (built-up areas, roads as well as the cat owners’ house) and (iii) green belts (i.e., large gardens or urban parks). All habitat use (based on the proportion of GPS points found in each habitat) and habitat availability (based on the proportion of area of each habitat available) was measured within 100% MCPs [11]. The ‘lost’ home points (when cats were known to be at home and some owners removed the collar) were added to the urban points calculated in ArcView to avoid underestimating the urban space usage. Lost points were based on the total hours at home and the mean number of waypoints generated/hour. This was necessary to avoid under-estimating the time spent at home. All such points were simply categorised as “urban”.

### 2.3. GPS Data

GPS data were imported into the Geographic Information Software (GIS) ArcView GIS 3.3 using @Trip (http://www.a-trip.com/download (accessed on 30 October 2009)) for analysis of spatial data. Preliminary tests of the GPS devices revealed a mean error of 30 m (S.D.) in residential (U) areas and thus we screened the data, removing any obvious outliers (i.e., >30 m between successive points) in all areas.

### 2.4. Statistical Analyses

Mann–Whitney U-tests were used to test for differences in the distance travelled per 24 h and home range size of cats between (i) U and UE; (ii) males and females; and (iii) summer and winter. To test for differences in the distances travelled by each cat during the day and at night, a Wilcoxon matched-pairs signed rank test was used. Spearman rank correlation analysis was used to test for associations between age and distance travelled and home range size. Differences in habitat use versus habitat availability were assessed using a ꭓ^2^ test.

## 3. Results

### 3.1. Home Ranges and Distances Travelled

The mean home range size (95% KDE) of UE cats (3.75 ± 0.95 ha) was not significantly different to the mean home range size of U cats (2.26 ± 0.78) in summer (Mann–Whitney U-test: U_7,7_ = 17, *p* = 0.37, Table 1). There was also no significant difference in the mean (±SE) distance travelled per day by U (9.1 ± 2.6 km) versus UE cats (14.5 ± 1.0 km) in summer (Mann–Whitney U-test; U_7,7_ = 16, *p* = 0.31). Winter samples were too small to differentiate home ranges of cats on the UE (*n* = 4) from U (*n* = 5).

### 3.2. Seasonal Differences—Summer vs. Winter

The mean home range size (95% KDE) of the cats in summer (3.00 ± 2.35 ha) was significantly larger than in winter (0.87 ± 0.25 ha; Mann–Whitney U-test: U_14,9_ = 16, *p* = 0.002; Table 1). The mean home range measured with minimum convex polygons (MCPs) was also significantly larger in summer (Mean = 31.65 ± 14.59 ha) than in winter (3.43 ± 1.24 ha; Mann–Whitney U-test: U_14,9_ = 8, *p =* 0.00016; Table 1).

The mean distance travelled per day (11.8 ± 5.8 km vs. 6.3 ± 3.1 km) and the maximum displacement from the home (528 ± 201 m vs. 184 ± 72 m) were both significantly greater in summer than winter: Mann–Whitney U-test: U_14,9_ = 28, *p* = 0.0275; Mann–Whitney U-test: U_14,9_ = 8, *p* = 0.00016, respectively (Appendix A).

The longest mean distance travelled per day (18.1 km) was by a male U cat and the shortest mean distance travelled per day (1.0 km) was by a female U cat (Appendix A). The sex of the cats had no significant impact on the mean distance travelled per day in summer (11.8 ± 5.8 km) Mann–Whitney U-test: U_6,8_ = 12, *p* = 0.14) and there was no correlation between the mean distance travelled daily in summer and the age of the cats (Spearman rank correlation analysis: Rs = −0.25, *p* = 0.38).

The mean maximum displacement from home (±SE) for U cats (466.9 ± 78.6 m) was about 20% lower than, but not significantly different from, the mean maximum displacement for UE cats (588.1 ± 71.2 m; Mann–Whitney U-test: U_11,12_ = 19, *p* = 0.52; Figure 3). For both seasons, the cat that travelled the furthest away from home (849 m) was a female UE cat roaming an urban green space. The shortest maximum displacement from home (73 m) was by a female U cat. The sex of the cats had no significant impact on the maximum displacement from home (393 ± 235 m) Mann–Whitney U-test: U_10,13_ = 22, *p* = 0.85) and there was no correlation between the mean maximum displacement from home and the age of the cats (Spearman rank correlation analysis: Rs = −0.21, *p* = 0.47).

Urban cats in summer travelled a mean distance (±SE) of 4.8 ± 1.5 km during the day and a mean distance of 4.3 ± 1.2 km during the night. UE cats travelled a mean distance of 8.6 ± 0.9 km during the day and 6.0 ± 0.7 km at night. Both U and UE cats travelled significantly more (Wilcoxon matched pairs test: T = 17, N = 14, *p* = 0.03) during the day than at night in summer. Samples were too small in winter for a similar analysis.

A higher proportion of the 78 cats (59%) returned prey to the home during summer compared to only 30% of the 27 in winter. Of the UE cats, 82% of the 11 entered TMNP, with a greater proportion doing so in summer (seven of seven) than in winter (two of the five). Only 31% (4 of 13) of the U cats living >150 m from the urban edge used natural green spaces. This suggests that there is some motivation to travel well away from the home to penetrate more natural habitats.

### 3.3. Movement in Relation to Available Habitats: Urban vs. Urban-Edge Area

As expected, cats living near the urban edge had a larger proportion of natural habitat (55.3 ± 4.1%) in their home range relative to U cats (2.6 ± 2.6%). UE cats also spent significantly (χ^2^ = 27.7, df = 1, *p* < 0.01) more time in natural habitat (17.0 ± 7.1%) than U cats (0.1 ± 0.1%), but cats in both areas used urban habitat more than expected (χ^2^ = 45, df = 1, *p* < 0.01).

## 4. Discussion

Our short-term GPS-tracking study of cats in an African city, the first for owned domestic cats on the continent, was designed to give a first-order assessment of how free-roaming domestic cats in a biodiverse African landscape use available habitats in two contrasting seasons, and by day and night.

The mean home range size based on the 95% kernel for the 22 cats tracked in this study (i.e., both seasons) was 3.0 ha—smaller than the estimate for feral cats in an urban mosaic in another South African study (9.3 ha [25]) and other areas [28], but similar to other studies of owned domestic cats in cities around the world (i.e., 2 to 4 ha [3]). However, home range size in this study varied markedly, being three-fold larger in summer than winter (3.00 vs. 0.87 ha, respectively). Since all cats were neutered and had access to natural habitat, and neither sex nor age influenced the home range size, it seems most likely that the cold and particularly the wet weather limits the time cats spent outside the home in winter.

Could the reduced number of study areas in winter (two suburbs) have influenced the reduced foraging return and smaller home range size of cats in the winter? The two suburbs sampled in winter (and also sampled in summer: Figure 2) were little different from those used in the summer study and offered two protected green spaces for cats to access (TMNP on one side and Glencairn Wetland Reserve on the other). That the proportion of cats returning prey (30% in winter versus 59% in summer) and the home range size were both smaller in an area with more protected areas in which to hunt suggests that our results (of reduced hunting in winter) were unrelated to habitat or opportunities. While the presence of naïve prey may induce cats to hunt more in spring, this is unlikely to explain the higher prey returns in summer because Cape Town is a winter-rainfall area and bird and mammal prey are stimulated to breed by rainfall, not temperature. Thus, fewer vulnerable birds and mammals are likely to be present in the hot, dry summer. The cold, wet weather remains the best explanation for the low proportion of cats hunting and the smaller home ranges in winter.

That fewer cats returned prey items home, and venture less far in winter, indicates that “catscapes” are smaller and pose less risk to wildlife in cool, wet weather. This accords with findings in a year-long study of domestic cats [9] in temperate North America indicating that 85% of prey captures recorded by KittyCams were caught in the warm boreal summer months (March–November). Similar trends were apparent in the KittyCam study of stray cats by Hernandez [30]. Together, these findings highlight the importance of considering season when extrapolating from seasonally constrained studies (e.g., [10] to annual and global cat predation estimates [3].

Significantly, from a conservation perspective, 9 of 11 UE cats in this study ventured into the protected areas of either the Table Mountain National Park or urban nature reserves within the cadastral boundaries of the city (Figure 2). Of concern is that 25% of the 12 U cats (i.e., those housed >150 m from the closest urban edge) also ventured into these protected areas, with implications for the size of any cat-free zones that may be proposed to help protect biodiversity [31]. Based on the average displacement of 588 m for the UE cats in summer, we recommend domestic cat-free buffers to span at least 600 m from any protected area boundary. This would not prevent feral cat intrusions, however, which seem to have larger home ranges across the globe (see Table 1). Some feral cats may also live and forage permanently within protected areas [32,33]. This effectively makes all species recorded as prey of cats vulnerable to the unnaturally high predator density of domestic cats near such parks.

Testimony to Cape Town’s cats wide-ranging behaviour is the very high MCP-derived mean home range of 31.7 ha compared to other MCP means for domestic cats around the world, including Denmark (5.78 ha), New Zealand (1.8–3.2 ha) and Australia (2.92 ha; Table 1). This implies that while the core areas of use are broadly similar for domestic cats worldwide, some Cape Town cats may do more short-duration, but greater-distance, excursions from their homes. Thus, while both U and UE cats used urban areas more than expected based on the relative availability of this habitat type alone, they may still impact wildlife in protected areas during forays deep into protected areas.

It is possible that caracal (*Caracal caracal*), a medium-sized felid that regularly patrols Cape Town’s urban edge [34] and readily includes domestic cats in its diet [35,36], increases the risks to domestic cats of spending time in the park. There are no other wild mammalian predators that regularly take cats around Cape Town, and predation of cats by snakes is unreported. There are no records of raptors in the region (e.g., Verreaux’s Eagles, Fish Eagles) depredating cats in the Cape Peninsula. Known predators of cats such Crowned Eagles *Stephanoeatus cornotus* and Leopards *Panthera pardus* [37] are not found in Cape Town. Comparable risk and avoidance effects have been noted in parks in the USA where Coyote *Canis latrans* presence limited the time cats spend in the open [38] and cat abundance was negatively associated with the presence of coyotes in Canada [39].

## 5. Conclusions

Domestic cats in Cape Town are not restricted to their owners’ property and range extensively in both urban and natural habitat, travelling as much as 18 km in a 24 h day and roaming over 800 m into protected green spaces. The mean home range size was similar to that of domestic cats in other cities worldwide but varied markedly with season. The warm summer months see approximately twice as many cats (30% vs. 59%) returning prey home than in winter, covering greater distances, moving further from their homes and regularly entering protected areas [8]. Given that cats exact a massive toll on native animals, mitigation measures to protect biodiversity in protected areas are essential, and our data on movements suggest cat-free buffers of at least 600 m may limit domestic cat intrusion outside the cadastral boundaries on Cape Town and into the species-rich Table Mountain National Park. Keeping cats indoors, particularly in summer, limiting their roaming with “catios” and adding bells and bright ruffs on collars can also reduce the predation on native prey [40,41]. The most persuasive argument for cat owners to keep their cats inside may be the dangers posed by meso-predators, cars and pathogens [42]. This may never be truer than in an African setting, with its high complement of predators and diseases.

## Figures and Tables

**Figure 1 animals-13-01013-f001:**
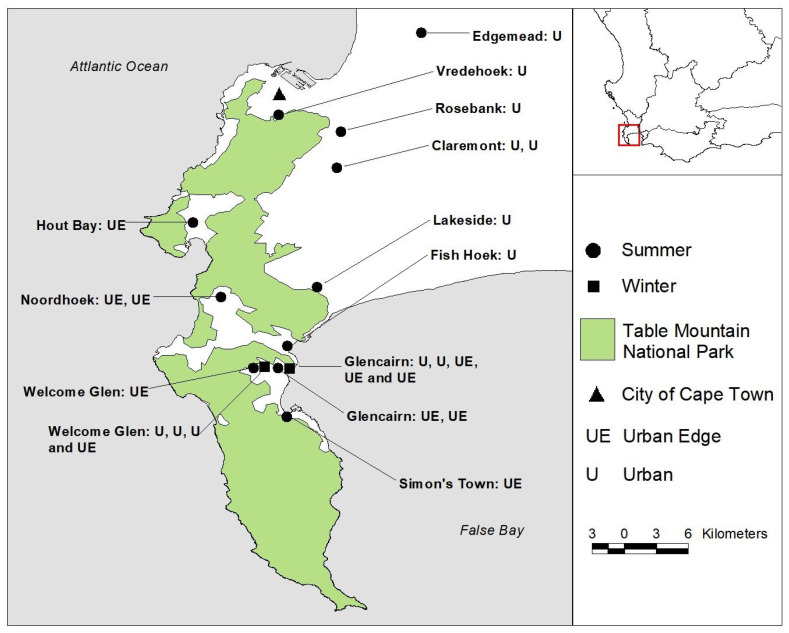
The Cape Peninsula study area, South Africa, showing the extent of Table Mountain National Park (green polygon) and the distribution of 14 summer-tracked cats in Cape Town suburbs and nine winter-tracked cats focused on the Glencairn/Welcome Glen area. Urban-edge (UE) and urban (U) cats are distinguished.

**Figure 2 animals-13-01013-f002:**
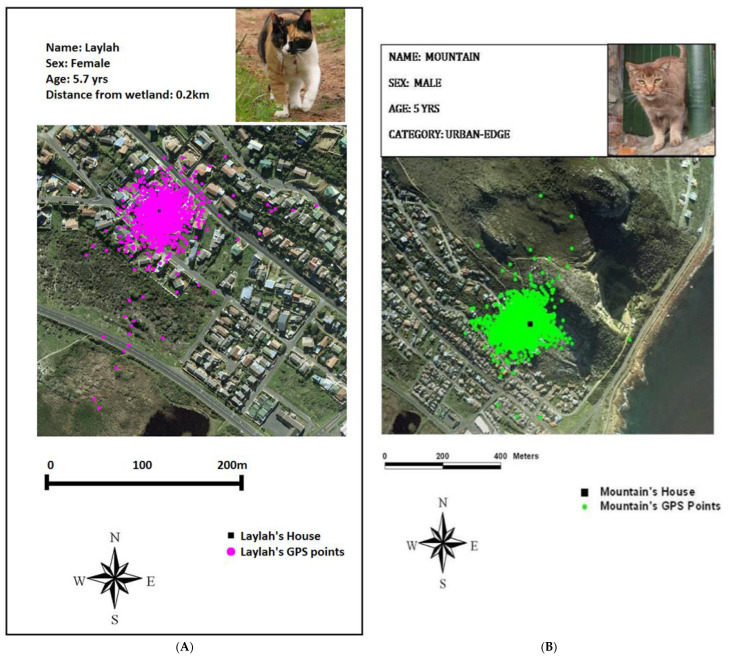
An example of the larger summer (7-day) home ranges of cats compared to winter ranges for UE cats adjacent to the same urban natural areas (Glencairn wetland and TMNP). The winter track (**A**) of Laylah, a female housed close to the wetland, with a 95% kernel range of 1.03 ha and the largest displacement of any cat in winter of 298 m. The summer track of Mountain (**B**), with a home range more than twice the size at 2.71 ha and maximum displacement about twice that of Laylah (of 567 m). Note the difference in scales.

**Figure 3 animals-13-01013-f003:**
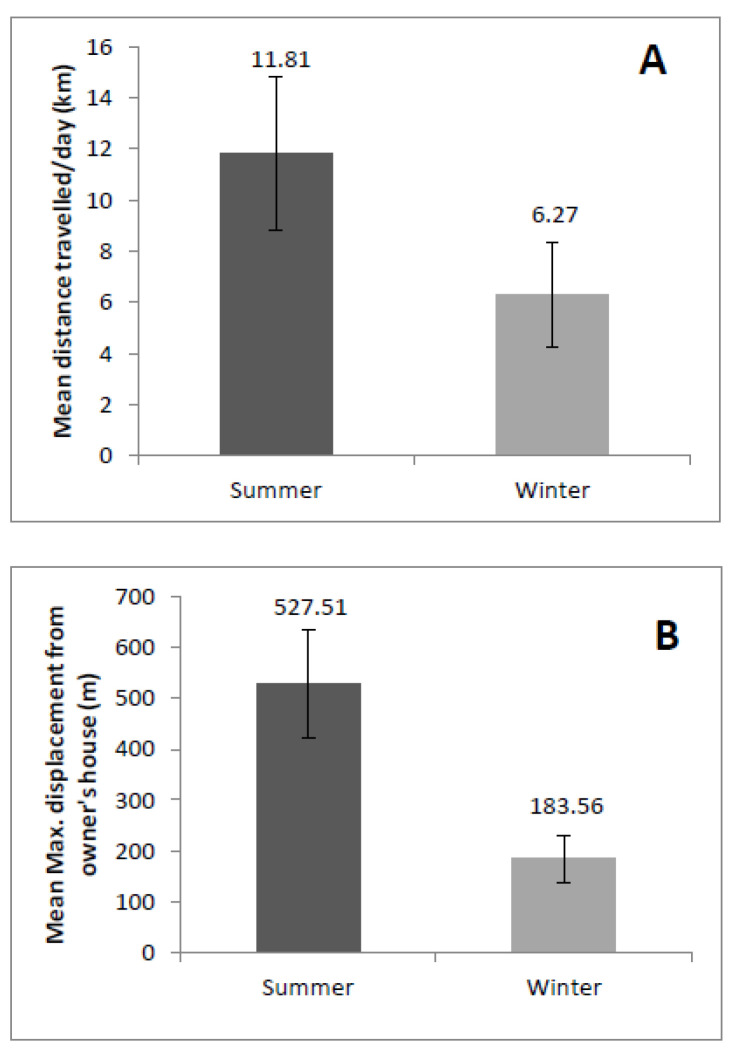
A seasonal comparison (winter versus summer) of (**A**) the average difference in mean distance travelled per day by domestic cats and (**B**) the maximum displacement from home.

**Table 1 animals-13-01013-t001:** Comparison of the mean home range area (95% kernel density estimates, KDE) of adult free-roaming domestic cats (owned and feral) across published studies. Early studies only reported maximum convex polygon (MCP) as a measure of home range, so we have also included that metric. Urban (U: > 150 m from the urban-edge) and urban-edge (UE: < 150 m from the urban-edge) cats are distinguished in our study and that of Pririe et al. [24], who used different distances (1000 m) for Non-Boundary vs. Boundary cats.

Source (Owned or Feral)	Location	Duration (Season)	Mean Home Range Area (KDE 95% (ha))	Mean Home Range Area MCP (ha)
This study (owned)	Cape Town, South Africa	5–10 days (summer)	2.26 ± 0.78 (U) 3.00 ± 2.35 (both U and UE) 3.75 ± 0.95 (UE)	31.65 ± 14.59
This study (owned)	Cape Town, South Africa	7–10 days (winter)	0.87 ± 0.25 ha	3.44 ± 1.24
Pillay et al. [25] (feral)	Kwazulu Natal, South Africa	11 months (all year)	9.31 ± 7.54	6.19
van Heezik et al. [14] (owned)	Dunedin, New Zealand	6 days (not reported)	Not reported	3.2
Meek, [22] (owned)	Sydney, Australia	7 days	Not reported	2.92
Horn et al. [26] (owned and feral)	Illinois, USA	Year round	5.16 ± 4.89 male (owned) 1.95 ± 0.87 female (owned) 103.17 ± 73.52 male (feral) 57.92 ± 33.61 female (feral)	1.83 ± 1.42 male (owned) 1.92 ± 1.09 female (owned) 157.01 ± 89.44 male (feral) 56.59 ± 21.34 female (feral)
Morgan et al. [27] (owned)	South Island, New Zealand	12 months	Not reported	1.8
Kays et al. [3] (owned)	Australia, New Zealand, USA, UK, Denmark	5 to 20 days (not reported)	2.5 ± 3.9 (Australia) 4.3 ± 6.8 (New Zealand) 4.6 ± 7.3 (UK) 4.7 ± 6.5 (USA) 3.6 ± 5.6 Overall mean	Not reported
Bishoff et al. [10] (owned)	Norway	30 days (Spring)	2.6 ha (range 0.3 to 21.0 ha)	5.78 ± 19.78
Zhang et al. [28] (feral)	China	Year round	12.6± 2.6 (male, breeding) 6.68 ± 1.22 (female, breeding)	Not reported
Thomas et al. [29] Pririe et al. [24]	United Kingdom	Year round	3.42 ± 0.61 (Boundary) ^1^ 2.01 ± 0.70 (Non-boundary) ^2^	6.88 (range: 1.69–11.5)

^1^ “Boundary” means within 100 m of the urban boundary. ^2^ “Non-boundary” means more than 1000 m from the urban boundary.

## Data Availability

Not applicable.

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
