# Peer review of "Seasonal Movement Patterns of Urban Domestic Cats Living on the Edge in an African City"

_animals, 2023, doi:10.3390/ani13061013_

Round 1

Reviewer 1 Report

Review of “Domestic cats on the urban edge roam further and more frequently etc….”

This study investigates movements of 23 domestic cats, using GPS, in Cape Town, South Africa, and describes home range, movements, and broad-scale habitat use during two seasons, comparing two groups of cats, defined as deep-urban (>150, from a green space) and urban-edge (<150m from a green space). It is the first GPS-tracking study in Africa. The manuscript is well-written and mostly clear, and presentz interesting and useful data.

Points to consider are:

I don’t think that a cat that lives only 150m away from a green space should be called “deep-urban”, because 150m is less than the distance across many cats’ home ranges. Therefore these “deep-urban” cats would have access to green spaces. On the map in figure 1 all the locations of the cats look like they’re on the urban edge.

Page 3. Various place names are given and Figure 1 is referred to, but the place names are not on the figure (Glencairn heights, Hout Bay, Welcome Glen). If these locations are referred to by name they need to be added to the map.

  I find the figure and text in the surrounding section confusing. The figure legend says the cats labelled are those tracked during the summer study (14), but then it mentions the winter study when 9 cats were tracked, and Mountain and Truffle are mentioned. I assume these two cats weren’t in the winter study? Why do the summer cats get shown on a map but the winter cats not? Perhaps this figure should show two maps, one for summer cats and one for winter cats.

Line 116: why have the mean ages of the winter cats not been presented for both winter and summer cats?

Line 120. How long were cat owners required to record all prey items and how was the information communicated and collated? This might have been described in a thesis or another paper, but it also needs to be reported here. The authors could make it clear here that a more detailed analysis of these data has been made in another paper.

Lines 157-158. I don’t think it is adequate to assume that 5-6 days of tracking is enough for home ranges to have reached an asymptote. The paper by van Heezik et al. reports that home ranges of several cats in their study did not reach an asymptote. An incremental area analysis needs to be done to show that the home ranges in this study were fully revealed after seven days. I they weren’t, then the authors should justify their inclusion in this study.

Line 187: “were”, not “was”.

Lines 189-190; Good to measure mean error: were the devices tested in the same habitats the cats were being tracked in? And how was that error (30m) taken into consideration when looking at habitat use?

Line 198. Using a chi squared test is a crude approach – possibly a better analysis would be to calculate a resource selection function with the variables of habitat, sex, age and whether edge or deep urban.

Lines 202-206. Here it is clear that separating them into edge and deep isn’t really warranted, when cats are covering at least 2.26 ha over 7 days and travelling 9km and 14km in a day.

Lines 206-207. On the previous page it is written that there were 4 and 5 cats, not 6 and 3.

Table 1. I’m not convinced of the value of including feral cats in this table. They are very different as are not food subsidised and this study’s aim isn’t about comparing the two types.

Figure 2. Laylah spelled differently in figure to figure legend. According to your definition Laylah is a deep-urban cat, but is clearly not looking at this aerial photo as she is situated close to a green space.

Lines 228-229. Don’t use “longest” and “lowest” together. Use either “longest” and “shortest” or “lowest” and “highest”.

Line 252: “cats” not “cates”

Lines 277-278. Is it possible that there are just more prey around in spring and summer as well, in that there will be young birds and reptiles that are newly independent and more vulnerable to predation. This could be a motivation to go out and hunt rather than just nicer weather.

Lines 288 – 289. Again, this is hardly surprising given what is known about the extent of cat movements.

Discussion of buffer size could acknowledge the studies of Lilith and Metsers et al. (2010) who also calculated required buffer sizes based on their tracking data.

Line 299. Was this value from one individual? Cats are well known to be highly variable in their home range size. In the van Heezik et al. study one individual living on the urban edge had a home range of 28 ha after only 3 days of tracking. This is worth mentioning. It means that the value for Cape Town isn’t necessarily higher than values obtained elsewhere.

Lines 301-302. I don’t think this suggestion is justified, as it depends on the detail of home range sizes reported by other authors. Outliers like that can be hidden in mean or median values.

Reviewer 2 Report

I found this to be an interesting and generally well executed study on domestic cats in Cape Town. The impact of cats might be less significant than in Australia and New Zealand, but it would be still a factor to consider even in Africa with its multitude of native predators. However, I assume that restrictions on cat ownership will be as difficult to enforce as elsewhere.

There is, however, one point I feel uneasy about and that is the measure of habitat preference. I gather that habitat availability was assessed only within the 100% MCP of a specific cat. Any habitat preference that might have determined the placement of the home range within the wider environment was therefore not assessed. Furthermore, an MCP is a non-statistical, unweighted measure of a home range. In contrast, the GPS locations are clustered around the activity centre / home of the pet cat that is by default within the urban setting. Combining these two measures has to provide a biased result for a preference for an urban habitat. Perhaps the home range centre should be excised from the analyses to avoid the bias.

Specific comments:

L71      Could the authors elaborate on what type of buffer they consider; banning cat ownership in the buffer zone or any other form of restrictions that might limit roaming.

L73      Here and actually throughout the introduction it is not clear whether the authors refer to the species in general or to owned cats. Initially, the term ‘domestic cat’ was used as a reference to the species especially when referring to predation rates in Australia with large feral cat populations. Here, however, I think the authors refer to animals owned and cared for by human owners. I think it is prudent to distinguish between domesticated, owned cats and their feral, wild counterparts.

L87      What was the rational for selecting this 150m threshold?

L149    I cannot find the van Heezik et al., 2010 reference in the literature list. I assume the authors refer to domestic cats only.

L190    How did this error imping on assessing habitat preference? I assume a number of points would have been close to boundaries between habitat categories. Were these excluded?

L198    Since the homes of all these cats were in an urban setting, I would assume that many if not most GPS fixes represent the area close to home. Therefore, a simple analysis based on the number of points inside and outside urban areas appears to be flawed as the MCP does not include any measure for the location of the home range centre. I would suggest excluding the 50% core area from the analysis thereby focusing on excursions.

L209/Table1    The explanation for (boundary, non-boundary) for the last to studies listed is missing from the table heading.

L214    The authors mentioned earlier that the areas cats were recruited from differed somewhat between summer and winter with winter cats coming from a more restricted area. I therefore wonder whether this statistical difference between seasons could also be explained by differences in the location of the recruited cats. At least this possibility should be discussed.

L250    Was there any indication that the movement of cats was restricted by the owners? The authors mentioned earlier that some owners removed the collar to charge it overnight.

L251    Were these prey items actually native animals or perhaps included animals that could have been caught in an urban setting such as introduced mice and rats?

L274    I thought all cats were sterilized – then why a reference to fertility status?

L293    I cannot follow this argument. A buffer zone excluding domestic cats would have no bearing on feral cats.

L296    With many species of native predators present in Africa the statement suggesting that all prey animals are protected when cats are excluded appears to be a bit strong. Please reword.

L307    Why this focus on a single native predator? I would have thought there are a few other species cats should be concerned about such as eagles and snakes. There are claims that the odd leopard around too.

L318    I am a bit confused here. The 30% and 59% figures were used in relation to prey being returned by cats in winter and summer, respectively. How do these suddenly translate into incursion rates?

 Supplementary Table 2         Should the last row be deleted?

Reviewer 3 Report

General Comments

This is an interesting manuscript and certainly has implications for management of domestic cats in urban areas. However, there are numerous errors both of omission and formatting that will require considerable effort to correct.

1)      Consistency in either having a space/or not between numbers and units of measure throughout the document.

2)      It is usual practice to separate cited references by a semi-colon rather than a comma.

3)      Consistency in either placing a comma after the cited reference or omit.

4)      et al. should either be in italic or not throughout the document.

5)      Numbers less than 10 should be spelled e.g., nine cats rather than 9 cats. For example, lines 105, 109 and 117 etc.

6)      Figure 2 is not referenced in the main text. Also, not sure of its significance given the different genders of the two cats.

7)      Why are the References numbered?

8)      There is absolutely no consistency in how the References are written/presented and this needs a thorough review.

9)      van Heezik et al. 2010; Conservation International 2008; Meek 2003 and Morgan et al. 2009 are cited in the main text but are not listed in the References.

Specific Comments

Line 34 – should be ‘its’ rather than ‘it’

Line 35 – ‘study’ rather than ‘studies’

Line 36 – Twenty-three cats rather than a total of 23

Line 43 – scientific name for caracal

Lines 50, 51 and throughout document – for consistency, cited references should either be listed alphabetically or chronologically

Line 54 – do not rather than don’t

Lines 56, 122 – et al should have a full stop after al.

Line 93 – not sure what has happened to the degree symbol?

Line 103 – where is the light green polygon?

Lines 112and 125 – Glencairn, Welcome Glen, Glencairn Heights and Hout Bay (not shown in Figure 1 as suggested)

Line 117 – space required between +

Line 138 - space required between (Coughlin and van Heezik 2015).This was

Line 149 – for consistency et al. should not be italic and no comma before date of publication

Lines 260 and 262 – Chi2 symbols are different, and the probability sign is now a capital (P) compared to earlier probability signs (p)

Round 2

Reviewer 1 Report

I appreciate the revisions the authors have made to this paper. There are a couple of points I would like to see addressed.

The authors have justified categorising cats as “deep urban” or “urban edge” when most cats are pretty close to an edge or a green space. Lines 86-90 says “When this study was initiated, there were no published GPS-data on cat distances. We, therefore, selected a figure of 150m as the threshold between DU and urban-edge cats based on our knowledge of cat behaviour. This is similar to a 98 m radius circle representing a circular 3-ha home range and thus biologically meaningful in hindsight. It also allows us to determine the distance a DU cat is prepared to go to penetrate a”

Unless this study was initiated before 2010, which seems unlikely, there were a couple of published accounts based on GPS data on distances moved by domestic cats – Metsers et al. (2010) and Van Heezik et al. (2010). Metsers et al. found maximum distances moved from the home were between 200m and 2.3km. In the other paper home ranges are reported but distances moved can be inferred. The explanation provided here by the authors of a 98m radius based on a 3ha home range is plausible – I guess I just object to the use of the term “deep urban”, which implies a location much more embedded in an urban matrix. When looking at the map some locations labelled as deep urban just don’t seem to be “deep urban”; e.g., Glencairn, Welcome Glen, Vredehoek.

Could the authors please consider the use of another term. Also, the sentence they have inserted in red is unfinished.

Line 160: In fact the Van Heezik et al. study acknowledge that for a number of cats 5 or 6 days was not sufficient to reveal the home range fully. The authorts should acknowledge in the discussion that it is possible that some of these home range estimates may be under-estimates.

Line 214-215 – table footnote. Here >1000m is defined as “non-boundary”  and similar to their differentiation of urban edge and deep urban, but deep urban is defined in this study as >150m, not 1,000m.
